# Shape Equivariant Learning for Robust MRI Segmentation

**Ainkaran Santhirasekaram**[1]                                     A.SANTHIRASEKARAM19@IMPERIAL.AC.UK
**Mathias Winkler**[2]                                              M.WINKLER@IMPERIAL.AC.UK
**Andrea Rockall**[2]                                               A.ROCKALL@IMPERIAL.AC.UK
**Ben Glocker**[1]                                                  B.GLOCKER@IMPERIAL.AC.UK
[1] *Department of Computing, Imperial College London, United Kingdom*
[2] *Department of Surgery and Cancer, Imperial College London, United Kingdom*

**Editors:** Under Review for MIDL 2023

## Abstract

The reliability of deep learning based segmentation models is essential to the safe translation of these models into clinical practise. Unfortunately, these models are sensitive to distributional shifts. This is particularly notable in MRI, where there is a large variation of acquisition protocols across different domains leading to varying textural profiles. We hypothesise that the constrained anatomical variability across subjects can be leveraged to discretize the latent space to a dictionary of shape components. We achieve this by using multiple MRI sequences to learn texture invariant and shape equivariant features which are used to construct a shape dictionary using vector quantisation. This dictionary is then sampled to compose the segmentation output. Our method achieves SOTA performance in the task of single domain generalisation (SDG) for prostate zonal segmentation.

**Keywords:** Shape Equivariance, Robustness, Segmentation, MRI

## 1. Introduction

Magnetic resonance imaging involves a complex acquisition process which differs across subjects and domains. This can lead to varying textural profiles and artefacts. Deep learning based segmentation models are however not robust to textural shifts and unencountered artefacts at test time. Domain generalisability for deep learning has been traditionally tackled through augmentation based strategies such as CutOut (DeVries and Taylor, 2017) and BigAug (Zhang et al., 2020). AdvBias (Chen et al., 2020) is an adversarial technique for MRI data which learns to generate bias field deformations to improve model robustness for segmentation. RandConv (Xu et al., 2020) which is perhaps the most related work, attempts to learn textural invariant features by using a randomised convolutional input layer. Here, we propose an alternative method to learn shape equivariant features based on the principle that in MRI, T2 weighted images and ADC maps calculated from diffusion weighted imaging contain the exact same spatial information and only differ in their textural profiles. There is anatomical consistency across subjects meaning there is reduced spatial variation in the segmentation outputs. Therefore, we propose to constrain the latent space to a dictionary of shape components which is sampled to construct the segmentation output. We hypothesise this will improve the generalisability of any segmentation model which maps the input space, $\mathcal{X}$ to a lower dimensional embedding space, $\mathcal{E}$ using an encoder, $\Phi_e$ before mapping to the segmentation output, $\mathcal{Y}$ with a decoder, $\Phi_d$. This is achieved using vector quantisation (Van Den Oord et al., 2017) of the shape equivariant features to create a discrete shape

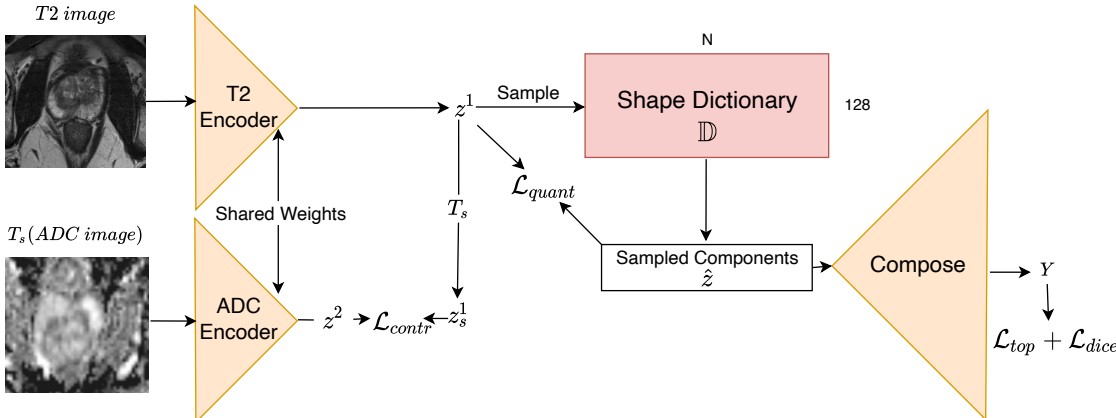

Figure 1: Overview of our method demonstrating using the ADC map to learn shape equivariant features which is quantised to construct a shape dictionary, $\mathbb{D}$.

space. We assume the dictionary is complete and sufficient to capture the entire distribution of segmentation outputs after composition of the discrete shape space using the decoder. We evaluate the capability of our method to improve domain generalisability in the task of prostate zonal segmentation with two labels (transitional and peripheral zone) when training on a single domain.

## 2. Method

We start with the image input which is the T2 weighted image, $x \in \mathbb{R}^{1 \times 256 \times 256 \times 24}$ and apply an intensity transformation, $T_i$ which is equivalent to acquiring the ADC map. We also apply a spatial transformation, $T_s$ to the ADC map which involves rotations. Specifically, we apply transformations from the dihedral group (d4) which consists of 90 degree rotation in the z plane and 180 degree rotation in the y plane. The order of this group is 8 so we create 8 transformations per sample during training. The T2 image and spatially transformed ADC map are passed through an encoder to produce their respective embeddings, $z^1$ and $z^2$ as shown in Figure 1. Shape equivariance and texture invariance is enforced by satisfying equation 1.

$$\Phi_e(T_s(T_i(x))) = T_s(z^1) \tag{1}$$

Therefore, we minimise the contrastive loss: $\mathcal{L}_{contr} = \|T_s(z^1) - z^2\|_2^2$. Note, a contrastive loss only theoretically learns equivariance to the 8 spatial transformations applied per sample. It does not constrain the convolutional layers to the D4 group. We assume an approximate equivariance to the D4 group by using our contrastive loss.

$$\mathcal{L}_{Quant} = \frac{1}{m} \sum_{i=0}^{i=m-1} \|sg(z_i^1) - e_k\|_2 + \beta\|z_i^1 - sg(e_k)\|_2 \tag{2}$$

We quantise, $z^1 \in \mathbb{R}^{128 \times 16 \times 16 \times 12}$ using vector quantisation by dividing $z^1$ into $16 \times 16 \times 12$ components and replacing each component in $z^1$ denoted $z_i^1$ with its nearest component,

| | Baseline | CutOut | BigAug | AdvBias | RandConv | Ours |
|---|---|---|---|---|---|---|
| Dice | 0.51±0.13 | 0.53±0.17 | 0.63±0.15 | 0.56±0.13 | 0.59±0.15 | **0.64±0.11** |
| HD | 0.40±0.11 | 0.37±0.19 | 0.25±0.12 | 0.33±0.15 | 0.29±0.08 | **0.23±0.10** |

Table 1: Dice score and Hausdorff distance(HD) ± standard deviation for different SDG methods compared to our approach

$e_k \in \mathbb{D}$ where $k = argmin_j \|z_i^1 - e_j\|_2$. This produces the discrete shape latent space, $\hat{z}$ which is inputted into the decoder to construct the segmentation output. The quantisation loss minimises the euclidean distance between $z_i^1$ and its nearest component, $e_k \in \mathbb{D}$ shown in equation 2. Stop gradients (sg) are applied to the correct operand. We compute the dice loss between the output, $\hat{y}$ and the T2 segmentation label, $y$. The total loss for training our framework is $\mathcal{L}_{total} = \mathcal{L}_{dice}(\hat{y}, y) + \mathcal{L}_{contr} + \mathcal{L}_{quant}$. Note, only T2 weighted images are required as input during inference.

## 3. Experiments and Results

**Dataset:** The training set comprises the Prostate dataset obtained from the Medical Segmentation Decathlon (Antonelli et al., 2022), consisting of 32 T2-weighted and ADC images captured at the Radboud University Nijmegen Medical Centre (RUNMC). We use the 30 T2 weighted images in the NCI-ISBI13 Challenge (Bloch et al., 2015) which was acquired from Boston Medical Centre (BMC) for our test set. All images are centre cropped to $256 \times 256 \times 24$ and normalised between 0 and 1.

**Baseline Model and Comparison:** We use a hybrid 2D/3D UNet as our baseline model in order to deal with the anisotropic Prostate MRI images. The encoder and decoder is made up of 5 levels consisting of 2D pre-activation residual blocks in the top 4 levels and a 3D pre-activation residual block in the bottleneck level. We use the same encoder and decoder architecture for our method. We compare our method to the following SDG methods applied to the baseline model: CutOut (DeVries and Taylor, 2017), BigAug (Zhang et al., 2020), AdvBias (Chen et al., 2020) and RandConv (Xu et al., 2020). All models were trained for up to 500 epochs using Adam optimisation with a learning rate of 0.001.

**Results and Discussion:** In Table 1, we show that our method outperforms other SDG methods in terms of the Dice score and Hausdorff distance. We therefore show that one can improve the domain generalisability of a segmentation model in an anatomical segmentation task by constraining the latent space to a finite set of shape components.

In future work, we will enforce D4 group equivariant convolutional layers by applying transformations from the D4 group to the filters themselves to create 8 transformed filters from each convolutional kernel. We will also constrain the convolutional kernels such that they are equivariant to other groups such as the SO(3) or SE(3) group as well as develop a method for SE(3) group equivariant vector quantisation.

## Acknowledgments

This work was supported and funded by Cancer Research UK (CRUK) (C309/A28804)

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
