# OpenReview forum: "Shape Equivariant Learning for Robust MRI Segmentation"
_MIDL.io/2023/Short_Paper_Track — MIDL 2023 Short paper track Poster_

### Official Review · Reviewer_gBae · 2023-04-24
**Review Of: Shape Equivariant Learning for Robust MRI Segmentation**

**Rating:** 6
**Confidence:** 4

**Review:**

The authors contrastively learn a spatial and intensity augmented discrete representation, which is then used for prostate MRI segmentation.

I think that this paper is lacking important details about its method. Throughout the abstract and section 1 (and the title) the authors talk about shape equivariance. However, in section 2 we learn that this equivariance is learned and not a constraint (i.e., we could learn non-equivariant features), and that it's encapsulated in $T_s$ a spatial transformation. How is $T_s$ parameterized? How is it generated? What is the expressive family of $T_s$? How does that family represent "shape" overall?

Further, this transformation must also be applicable to discrete codes $z$, which surely means some NN-type interpolation, which may not be equivariant.

Perhaps the authors misread the author instructions, since their submission is exactly 3 pages instead of 3 pages + references. This additional space should have been used in explaining $T_s$.

Overall I think there is something here worth discussing at MIDL, but I think explaining the main idea of the method is very important before any publication aside from an abstract.

---

### Official Review · Reviewer_duvm · 2023-04-24
**Shape equivariant learning to improve domain generalisability of segmentation**

**Rating:** 7
**Confidence:** 4

**Review:**

The goal of this work is  to improve the domain generalisability of a segmentation model by training it to learn texture invariant and shape equivariant features. The model is trained using multiple MRI contrasts (T2w and ADC map)  to learn shape equivariant features by constraining the latent space to a finite set of shape components using vector quantization (Van Den Oord et al., 2017) and to learn texture invariance by applying contrastive loss in feature space. The method is applied for prostate zonal segmentation with two labels (transitional and peripheral zone) for single domain generalization and achieved slightly improved results. Some limitations include: poor performance when using domain generalization (dice score of 0.64), limited novelty of the approach. Another point is that ADC maps from diffusion images may have different segmentation maps (due to nonlinear distortion artifacts) when compared to anatomical T2w images, and therefore it may be inaccurate to assume the same segmentation maps/ anatomical shape will apply to both modalities.